# Research Method of Discontinuous-Gait Image Recognition Based on Human Skeleton Keypoint Extraction

**DOI:** 10.3390/s23167274

**Published:** 2023-08-19

**Authors:** Kun Han, Xinyu Li

**Affiliations:** School of Traffic and Transportation Engineering, Central South University, Changsha 410075, China; 204211001@csu.edu.cn

**Keywords:** gait recognition, human keypoint detection, non-sequential image sequence, CSP structure, XBN block

## Abstract

As a biological characteristic, gait uses the posture characteristics of human walking for identification, which has the advantages of a long recognition distance and no requirement for the cooperation of subjects. This paper proposes a research method for recognising gait images at the frame level, even in cases of discontinuity, based on human keypoint extraction. In order to reduce the dependence of the network on the temporal characteristics of the image sequence during the training process, a discontinuous frame screening module is added to the front end of the gait feature extraction network, to restrict the image information input to the network. Gait feature extraction adds a cross-stage partial connection (CSP) structure to the spatial–temporal graph convolutional networks’ bottleneck structure in the ResGCN network, to effectively filter interference information. It also inserts XBNBlock, on the basis of the CSP structure, to reduce estimation caused by network layer deepening and small-batch-size training. The experimental results of our model on the gait dataset CASIA-B achieve an average recognition accuracy of 79.5%. The proposed method can also achieve 78.1% accuracy on the CASIA-B sample, after training with a limited number of image frames, which means that the model is more robust.

## 1. Introduction

Biometric technology utilizes the principles of biostatistics and combines them with modern technologies such as computers, optics, acoustics, and biosensors to use biometric information as a unique identifier for individual identification. It is currently widely applied in various areas of life, such as public safety and mobile payments. Biometric information can be divided into physiological information determined by an individual’s genes and behavioural information gradually formed by an individual’s experiences. Physiological information includes fingerprints [1], iris patterns [2], facial features [3], and DNA [4], while behavioural information includes human handwriting [5], gait [6], and so on.

Gait recognition is a new biometric recognition technology currently proposed, which aims to record and distinguish the movement trajectories of the trunk and limbs as the main identification information during the movement or walking of an individual [7]. Studies in related disciplines, such as medicine and psychology [8,9,10], have shown that human strength, tendon and bone length, bone density, visual acuity, coordination ability, experience, weight, centre of gravity, degree of muscle or bone damage, physiological conditions and personal walking style are maintained as an individual’s unique characteristics [11]; they are very distinctive, and these do not change easily.

Gait information can be captured by wearable sensors attached to the human body, including accelerometers [12], gyroscopes, pressure sensors [13], etc.; these can be analysed in series, using the principles of biostatistics. The non-wearable gait information acquisition method combines a computer and a camera; an ordinary 2 K camera is used to capture an uncontrolled pedestrian gait video of a crowd 50 m away in real-time. Further analysis is performed using computer technology. The process is called vision-based gait recognition. This identification method has the advantages of not requiring individual cooperation, and being difficult to deceive. It can also detect and measure in low resolution, allowing it to be used when facial or iris information cannot be recognized in high resolution. Visual gait recognition technology based on deep learning networks can autonomously extract gait motion features from image sequences without building a complex human body model, and its accuracy and robustness only depend on the design of the network structure and the training image library. It has promising applications in intelligent video surveillance, patient monitoring systems, virtual reality, intelligent security, athlete training assistance, and more.

The main characteristics of a pedestrian walking in a public place are small changes in the range of walking movements, but there are changeable walking angles and blocking by objects. In view of these characteristics, this paper proposes a research method of gait image recognition that leverages discontinuously the extraction of key points on the human skeleton.

The main contributions of our work are summarised below:(1)We attempt to extract gait features from discontinuous video images of walking, which have rarely been studied in gait recognition. We designed a discontinuous frame-level image extraction module, which filters and integrates image sequence information, and sends the relevant information obtained to the gait feature extraction network, to extract relevant features and identify pedestrians, improving the network’s gait characteristics for discontinuous time-series’ learning ability.(2)Compared with ResGCN, the improved gait recognition algorithm (based on ResGCN) uses a bottleneck for feature dimensionality reduction. It also reduces the number of layers of the feature map, reduces the amount of computation, increases the cross stage partial connection (CSP) structure and XBNBlock, and upgrades the spatial graph convolution (SCN) and temporal graph convolutional (TCN) network modules, with a bottleneck structure in the network. This maintains the same size of input and output, on the basis of reducing network parameters, so as to enhance the learning ability of the response module. It can also reduce memory consumption.(3)Our paper demonstrates the effectiveness of the improved network algorithm on the commonly used gait data set CASIA-B, and the recognition accuracy is better than other current model-based recognition methods, especially for pedestrian gait images influenced by a certain appearance. The effectiveness and generalization of the algorithm proposed in this paper were tested on both the CASIA-A dataset and a self-built small-scale real environment dataset.

The rest of the paper is organised as follows. Section 2 outlines the work related to this paper. In Section 3 and Section 4, we present our work in detail. Section 5 presents the experimental results and analysis, and Section 6 concludes the paper.

## 2. Related Works

### 2.1. Human Keypoint Detection Research

Human keypoint detection [14] mainly detects the head, shoulder, elbow and other joint parts of the human body in an image. It outputs the relevant parameters and positional relationship of all or partial parts of the human body through keypoints, to describe information regarding the human body skeleton. The position and category of joints are the basis of research fields such as human behaviour recognition and gait recognition. In computer vision tasks, research on human keypoint detection using convolutional networks is divided into single-person and multi-person keypoint detection. Our paper mainly considers the use of a single-point detection network to extract the skeleton keypoint coordinate information.

The testing methods can be divided into two categories:(1)Regression-based keypoint detection, i.e., learning information from input images to the keypoints of human bones through an end-to-end network framework. DeepPose [15] applied the Deep Neural Network (DNN) to the problem of human keypoint detection for the first time, and proposed a cascaded regressor that can directly predict the coordinates of keypoints, using a complete image and a seven-layer general convolutional DNN as input. The detection works well on series datasets, with obvious changes and occlusions. The authors of [16] proposed a structure-aware regression method based on ResNet-50, using body structure information to design bone-based representations. Dual-source deep convolutional neural networks (DS-CNN) [17] used a holistic view to learn the appearance of local parts of the human body, and to detect and localise joints. The authors of [18] proposed a multi-task framework for joint 2D and 3D recognition of human skeleton keypoint detection from video sequences of still images. In [19], a 2D human pose estimation keypoint regression based on the subspace attention module (SAMKR), was proposed. Each keypoint was divided into an independent regression branch and its feature map equally divided into a specified number of feature map subspaces, deriving different attention maps for each feature map subspace.(2)Detection based on the heat map detection method, i.e., the heat map of K keypoints of the human body is detected first, and the probability that the key point is located at (x, y) is then represented by the heat map pixel value *L_i_*(x, y). The heat map of bone keypoints is generated by a two-dimensional Gaussian model [20] centred on the location of the real keypoint, and the bone keypoint detection network is then trained by minimising the difference between the predicted heat map and the real heat map. Heatmap-based detection methods can preserve more spatial location information and provide richer supervision information. A hybrid architecture [21], consisting of deep convolutional networks and Markov random fields, can be used to detect human skeleton keypoints in monocular images. In [22], the image was discretised into a logarithmic polar coordinate region, centred on the skeletal keypoint, and a VGG network was employed to predict the category confidence of each paired skeletal keypoint. According to the relative confidence scores, the final heatmap for each keypoint was generated by a deconvolutional network. HRNet [23] proposed a high-resolution network structure. This model was the first multi-layer network structure to maintain the initial resolution of the image, while increasing the feature receptive field. This structure continuously expands the features through multi-stage cascaded residual modules. Receptive fields, multi-network branches to extract feature information with different resolutions, and multi-scale feature information fusion, make the network more accurate in detecting the keypoints of human skeletons. Since the heat map representation is more robust than the coordinate representation, more current research is based on the heat map representation.

### 2.2. Gait Recognition Research Methods

According to the type of gait feature extraction, the current gait recognition methods can be roughly divided into human appearance-based [24] and model-based [25].

Appearance-based Method. The binarisation algorithm is used to extract the contour in the image and the optical flow method is used to extract gait features such as speed, leg angle, and gait cycle time in the contour. In [26], the Gaussian mapping principle was used to calculate the curvature of the human body contour boundary in the binary image as the recognition feature. The authors in [27,28] extracted pedestrians’ walking and moving contours from the background and calculated the similarity of gait images by calculating the temporal changes in the contours to represent gait features. The other method is to use gait energy images (GEI) [29], chrono gait images (CGI), or frame difference energy image (FDEI), etc., to aggregate the contours of the complete gait sequence into an image template, and use a recursive learning strategy and a neural network to learn temporal features of the sequence for recognition. GaitNet [30] proposed an end-to-end gait recognition network that could be trained simultaneously, consisting of two convolutional neural networks for gait segmentation and classification. GaitSet [31] used a convolutional network to extract frame-level features of gait contours, using horizontal pyramid mapping (HMP) to map them to set-level features, and finally using Euclidean distance to perform the difference between the object to be identified and the template. Gaitpart [32] used a frame-level part-feature extractor (FPFE) and multiple parallel micro-motion capture modules (MCMs) to extract the spatio-temporal features of the contours of different parts of the human body. Ref. [33] used 3D convolutions and 3D local convolutions [34] to capture global or local spatio-temporal features under multiple views for gait recognition. The model is relatively difficult to train. This type of gait recognition method has the problem of losing time and fine-grained spatial information during the pre-processing process, and it is difficult to train the network model.

Model-based Method. Refs. [35,36] abstracted various parts of the human body into a stick model to simulate the movement characteristics of the human body while walking. In [37], the rotation angles of the hip and knee joints were used to characterise the gait when a person walks. Ref. [38] tracked and fitted the movement of the lower limb joints of the human body during walking, to obtain dynamic changes of the lower-limb-joint rotation angle over time, as the gait feature. Traditional skeleton modelling methods usually rely on hand-crafted features or traversal rules, resulting in limited expressive power and difficult generalisations. The human skeleton can also use the depth camera in ‘Microsoft Kinect’ to capture the action [39] and the segmentation strategy to separate the person from the background, analysing the image of the pedestrian in motion to obtain the various parts of the body and further extract relevant features. However, using the depth camera in Kinect has certain limitations on the shooting space and distance. At present, the deep convolutional neural network is more widely used to directly evaluate and extract the features of the human body structure in the image. In [40,41], OpenPose [42] was used to extract the keypoints of the human skeleton; the PTSN network was constructed to obtain the temporal and spatial variation characteristics of the joint sequence, to realise cross-view recognition. GaitGraph [43] proposed a gait graph that combines human keypoint detection with graph convolutional networks (GCNs), to obtain a model-based approach to end-to-end gait recognition. SDHF-GCN [44] handled skeleton-based gait recognition tasks via natural connections, temporal correlations, and symmetric interactions.

In the past, appearance-based gait recognition methods were more popular. However, advancements in human keypoint detection algorithms have brought about more development opportunities for model-based gait recognition methods.

## 3. Cross-Gait Recognition Network for Keypoint Extraction

We used Gaitgraph [43] as the original network, with the aim of improving the algorithm and realising the gait image recognition that leverages discontinuously the keypoint extraction of human bones. GaitGraph uses a graph convolutional network on the human skeleton pose map for gait recognition. Compared with the extraction of human appearance and outline, the human-skeletal-keypoint coordinate information and feature extraction only involve the 2D coordinates of the major joints of the body. Since the skeletal sequence only captures motion information, inferring the keypoint coordinates of the human body through model inference allows for the avoid the influence of contextual interferences such as background changes and lighting variations. The implementation process inputs a set of gait images for the model, uses YOLOv3 to detect pedestrian targets in the images, uses HRNet [23] to extract keypoints of the human skeleton, and then uses ResGCN to independently extract frame-level features from the pre-processed skeleton images.

We considered the design of the discontinuous frame extraction module of the gait video image after extracting the skeleton keypoint information, and selected the less-discontinuous image frame-level sequences (for example, 30 frames) from each group of skeleton keypoint information frames, to form a list. These were then sent to the gait feature network to extract gait-related features. The goal was to reduce the model’s reliance on temporal features of the images during the training process. The whole algorithm framework is shown in Figure 1.

### 3.1. Human Keypoint Extraction Network

We followed the HRNet used in GaitGraph as a 2D human keypoint extractor. In general, 2D poses have certain advantages in extracting keypoints of the human body. In practical applications, the amount of calculation required to extract two-dimensional keypoints is small, and the requirements for equipment performance are not very high. Compared to traditional serially connected network structures, HRNet connects high-resolution and low-resolution subnetworks in parallel, always maintaining high resolution, which enhances the accuracy of spatial prediction. To address the issue of multi-scale feature fusion, HRNet employs multiple instances of the multi-scale fusion module, utilizing low-resolution features with the same depth and similar level to enhance high-resolution features. This approach ensures both high resolution and sufficient feature representation during skeleton extraction.

HRNet uses a convolutional network to detect pedestrians from top to bottom in an image of dimensions W × H × 3, and then detects the positions of 17 key parts of the human body (such as elbows, wrists, etc.). Our chosen network consisted of two-stride convolutions, forming a backbone aimed at reducing resolution, a subject outputting with its input feature map, and a regressor estimating selected keypoint locations and converting them to a full-resolution rate heatmap. Specifically, HRNet-W48 consists of four stages and four parallel subnetworks; the resolution is gradually reduced by half, and the width (number of channels) is increased by two times. This means that the width of the high-resolution subnetwork in the last three stages (C) is 48, and the widths of the other three parallel subnetworks are 96, 192, and 384, respectively. A total of eight cross-resolution information exchanges were performed to achieve multi-scale fusion. The keypoint connection diagram of the human skeleton, extracted from the gait data set CASIA-B using HRNet, is shown in Figure 2.

### 3.2. Gait Feature Extraction and Recognition

The gait feature extraction network consists of ResGCN blocks. The ResGCN block consists of SCN and TCN, with a bottleneck structure and residual connections. The network sequentially consists of multiple ResGCN blocks, followed by an average pooling layer and a fully connected layer producing feature vectors, using supervised contrast (SupCon) [45] as the loss function.

(1)The spatial GCN operation of the graph convolutional network (GCN) for each frame *t* in the skeleton sequence is expressed as:

(1)$$f_{out}=\sum_{d=0}^{D}W_{d}f_{in}\left(\Lambda_{d}^{-\frac{1}{2}}A_{d}\Lambda_{d}^{-\frac{1}{2}}⊗M_{d}\right)$$ where *D* is the predefined maximum graph distance, *f_in_* and *f_out_* denote the input and output feature maps, ⊗ denotes element-wise multiplication, *A_d_* denotes the d-order adjacency-matrix-labelled joint pairs with graph distance d-th, and Λ*_d_* is used to normalise *A_d_*. Both *W_d_* and *M_d_* are learnable parameters, which are used to implement convolution operations and adjust the importance of each edge, respectively.

For temporal feature extraction, an *L* × 1 convolutional layer is designed to aggregate contextual features embedded in adjacent frames. *L* is a predefined hyperparameter which defines the length of the time window. The spatio-temporal convolution layer is followed by a batch normalisation (BN) layer and a ReLU layer, constructing a complete basic block.

(2)A residual connection with a bottleneck structure: The bottleneck structure involves inserting two 1 × 1 convolutional layers before and after the shared convolutional layer. In this paper, the bottleneck structure is used to replace a set of temporal and spatial convolutions in the ResGCN architecture. The structures of SCN and TCN with the bottleneck structure are shown in Figure 3 and Figure 4, respectively. The lower half of the structures in the figures is added to the upper half to achieve residual connections, allowing a reduction in the number of feature channels during convolution computations using a channel reduction rate *r*.

(3)The fully connected (FC) layer that generates feature vectors takes the residual matrix R as input. Each output of the FC layer corresponds to the attention weights of each GCN layer.(4)Supervised contrastive (SupCon) loss function: For the experimental dataset, the data is processed in batches. The pre-divided dataset is sequentially input to the training network, and two augmented copies of the batch data are obtained through data augmentation. Both augmented samples undergo forward propagation using the encoder network to obtain normalized embeddings of dimension 2048. During training, the feature representations are propagated forward through the projection network, and the supervised contrastive loss function is computed on the output of the projection network. The formula for calculating the SupCon loss is as follows:

(2)$${ℒ}_{out}^{\sup}=\sum_{i\in I}{ℒ}_{out,i}^{sup}=\sum_{i\in I}\frac{-1}{\left|P(i)\right|}\sum_{p\in P(i)}\log\frac{\exp(z_{i}\cdot z_{p}/\tau )}{\sum_{a\in A(i)}\exp(z_{i}\cdot z_{a}/\tau )}$$ where $P(i)\equiv \left\{p\in A(i):{\tilde{y}}_{p}={\tilde{y}}_{i}\right\}$ is the index set of all positive samples different from *i* in the multi-view batch, |P(i)| is the cardinality, and $\log({ℒ}_{out}^{\sup})$ is the sum of the number of positive samples.

We improved and optimised the ResGCN network structure, and its position is presented in Table 1.

#### 3.2.1. CSP Structure

CSP [46] structure can optimise the problem of repeated gradient information in the network; the structure is shown in Figure 5. The CSP structure of the SCN and TCN is optimised with a bottleneck structure in ResGCN, to achieve richer gradient combinations and reduce network calculations. The main idea of CSP is to spread the gradient flow through different network paths by splitting the gradient flow, and to further respect the variability of the gradient by integrating the feature maps at the beginning and end of the network stage. The specific method is to divide the feature map of the base layer into two parts. One branch passes through multiple residual structures, and the other branch directly performs convolution; the two branches performing are concatenated to realise that the characteristics of CSP are consistent with the number of output channels. The model learns more features.

#### 3.2.2. XBNBlock

Due to the stacking effect of batch normalisation (BN), the estimation bias is accumulated, affecting test performance. The limitation of BN is that the errors increase rapidly as the batch size becomes smaller. Batch-free normalisation (BFN) prevents the accumulation of such estimated offsets, hence the introduction of XBNBlock [47], a module that replaces a BN with a BFN in ResGCN, making it more robust to distributed offsets.

BFN avoids normalisation along the batch dimension, effectively avoiding the problem of statistic estimation. This paper replaces the second BN in the bottleneck layer with group normalisation (GN) [48]. The calculation of GN is independent of the batch size and can be used for tasks that occupy a large amount of video memory, such as image segmentation, in order to solve the problem of BN having a poor effect on small mini-batch sizes.

GN divides the channels into groups and computes, within each group, the mean and standard deviation (std) for normalisation.

The feature normalisation method performs the following computation:(3)$${\hat{x}}_{i}=\frac{1}{\sigma_{i}}\left(x_{i}-\mu_{i}\right)$$

Here, *x* is the feature computed by a layer and *i* is an index. In the case of 2D images, *i* = (*i_N_*, *i_C_*, *i_H_*, *i_W_*) is a 4D vector, indexing the features in (*N*, *C*, *H*, *W*) order, where *N* is the batch axis, *C* is the channel axis, and *H* and *W* are the spatial height and width axes, respectively.

The specific method is as follows: GN divides the channels of each sample feature map into *G* groups (each group will have *C*/*G* channels), and calculates the mean (*μ*) and standard deviation (*σ*) of the elements in these channels along the (*H*, *W*) axis and a set of *C*/*G* channels. Each group of channels is independently normalised with its corresponding normalisation parameter.

*μ* and *σ* in Equations (4) and (5) are computed by:(4)$$\mu_{i}\left(x\right)=\frac{1}{\left(C/G\right)}\sum_{c=gC/G}^{\left(g+1\right)C/G}\sum_{h=1}^{H}\sum_{w}^{W}x_{i}$$
(5)$$\sigma_{i}\left(x\right)=\sqrt{\frac{1}{\left(C/G\right)HW}{\sum_{c=gC/G}^{\left(g+1\right)}\sum_{h=1}^{H}\sum_{w=0}^{W}\left(x_{i}-\mu_{i}\left(x\right)\right)}^{2}+\varepsilon }$$

*G* is a pre-defined hyper-parameter (*G* = 32 by default), ε=1e−8.

The improved structures of TCN, SCN and ResGCN are shown in Figure 6, Figure 7 and Figure 8.

## 4. Experiments

This section provides a concise and precise description of the experimental results, their interpretation, and the experimental conclusions that can be drawn.

### 4.1. Computer Configuration

All experiments in this paper were conducted on a workstation with an NVIDIA RTX 2080Ti GPU. CUDA, cuDNN, OpenCv, and Pytorch were used to realize the detection model. The details of computer configuration are shown in Table 2.

### 4.2. Dataset and Training Details

CASIA-B [49] is a widely used large-scale multi-view single-person gait database. It was proposed by the Institute of Automation, Chinese Academy of Sciences in January 2005, according to the training and testing requirements of the gait recognition network. A total of 124 subjects, individual RGB images are provided, taking into account the three variations of viewing angle, clothing, and carrying-condition variations. The method uses 11 cameras to collect data from 11 viewing angles of each subject at an interval of 18°, from 0° to 180°, at the same time, and each subject in each viewing angle takes ten gait sequences; these include three different states, namely six gait sequences, NM#1–6, in the normal dressing state, two gait sequences, BG#1–2, with different backpacks, and two gait sequences with different coats, CL#1–2. Since the research for this data set focuses on the influence of factors other than human detection or segmentation, a simple background and indoor environment are used for video acquisition.

We selected CASIA-A [48], which can also provide RGB images, as a small-scale gait data test sample. It included walking videos of 20 people, and each person had 12 image sequences, containing 3 walking angles of view (0°, 45°, and 90°), with 4 image sequences for each direction.

At testing, the distance between the gallery and probe was defined as the Euclidean distance of the corresponding feature vectors. Besides this, we fed the original and a flipped order sequence into the network, and took the average of two feature vectors.

## 5. Experimental Results and Discussion

### 5.1. Basic Experiment

Figure 9 shows that, under the same training strategy, our modified model outperforms the baseline in both recognition accuracy and loss curve. The improved model has a more stable rise in recognition accuracy during testing. After training for 400 Epochs, the loss curve no longer declines significantly, and our model is more robust.

Table 3 shows the comparison between the improved-gait-recognition algorithm network and the baseline. For the photographed pedestrians walking normally, the network before and after the improvement has its own advantages and disadvantages in the recognition process of each angle, and the final average detection accuracy rate increases by 2.3%. For the photographed pedestrian carrying a backpacks and wearing a jacket, the improved gait recognition network in this paper shows better improvement in the recognition accuracy of each angle; its average accuracy is increased by 3.1% and 4.2%, respectively. Specific to each angle, in the case of the lowest recognition accuracy of 180°, the recognition accuracy of the photographed pedestrians carrying backpacks and wearing coats also increased, by 2.4% and 10.5%, respectively. This indicates that, after adding the CSP structure and XBNBlock, the extraction of gait features is more accurate, which effectively improves the defects of the original network in relation to identification in these two cases. The last row in the table shows the recognition accuracy after training on non-continuous frame-level images. Compared with the baseline, the final average detection accuracy of the photographed pedestrians walking normally is increased by 0.5%. For the photographed pedestrians carrying backpacks and wearing coats, the average recognition accuracy of 11 different viewing angles increased by 1.8% and 3.3%, respectively, indicating the effectiveness of the improved network in reducing the dependence on temporal features.

In order to prove the contribution of the three modules added in the improved model and to select the most suitable statistical function for image dimensionality reduction, an ablation experiment was designed, and the experimental results are shown in Table 4. It can be seen from the table that, after increasing the modules sequentially, the recognition accuracy gradually increases.

The comparison of recognition accuracy of eleven walking angles in three different walking states in the ablation experiment is shown in Figure 10. The three figures show that, whether or not the input gait image is restricted to discontinuous frames, the overall recognition accuracy of the improved network is higher than that of the baseline network. Figure 10b is in the knapsack state, and the improvement in recognition is more obvious. As can be seen from the line graph of the comparison of the state of wearing a jacket in Figure 10c, adding different modules has a greater impact on the recognition of different angles.

### 5.2. Multipart Figures

At present, there are many gait recognition algorithms based on the CASIA-B dataset. The current best-performing model is based on contour features. This paper compares both appearance-based and model-based methods with this improved model; the specific comparison values are shown in Table 5.

3DLocal is the best gait-image-recognition algorithm based on human appearance contours in the table, with an average recognition accuracy of 88.8%. In the gait recognition network based on human skeleton keypoints, the best recognition accuracy of the improved network in this paper is 79.5%. Although there is a certain gap between the accuracy of the algorithm based on human appearance contour recognition, it is worth noting that the accuracy of the skeleton keypoint information and discontinuous frame-level image information used in this paper can be the same as that of the GaitNet network recognition.

Table 6 uses the average accuracy rate, detection and recognition time per frame and file weight as network performance evaluation parameters, and compares the data of the algorithms after adding each network module. It can be seen from the table that, when the improved algorithm model does not limit the number of frames extracted for image training, the average recognition accuracy improves overall. The average detection time can be shortened by up to 0.0346 ms, and the average detection time weight of the improved model file size is reduced by 0.3 MB. After limiting the number of input image frame series during training, the average detection accuracy of the gait-recognition-algorithm models, before and after the improvement, decreases, and the average detection time increases; the file size weight did not change significantly.

### 5.3. Tests in CASIA-A

The experimental results of the test in CASIA-A are shown in Table 7. We followed the same training and testing protocol as the baseline model GaitGraph [43] method for fair comparison, and also tested discontinuous gait frame-level images. From the test results, it can be observed that our improved model achieves higher recognition accuracy than the baseline model across all three angles. The average recognition accuracy of the improved model is 5.9% higher than that of the baseline model. Even in the case of testing with non-consecutive frame-level gait images, the average recognition accuracy can still reach 89%.

### 5.4. Real-Scene Gait Image Test

In order to further verify the effectiveness of the algorithm proposed in this paper, we collected gait images in real scenes. We collected three walking states of six volunteers: normal walking (NM), backpack walking (BG) and wearing a coat (CL). Three walking angles of view (36°, 108° and 162°) and two image sequences for each state (#1–2) were used. Some of the data images are shown in Figure 11. In the test, NM#1–2 is used as the gallery and BG#1–2 and CL#1–2 are used as probes; the experimental results are shown in Table 8.

Since the image data of the self-built gait dataset is relatively limited, the data range for testing is narrowed down, so the recognition accuracy is slightly better than the test results in the public dataset. From the test results in Table 8, it can be observed that the average recognition accuracy of the algorithm in real-gait images reaches 82.3%, which is a good performance. When limiting the number of input image frames during the test, the average recognition accuracy is 79%. The overall recognition accuracy is not significantly affected by the reduction in the number of image frames, especially for the state recognition effects of pedestrians carrying backpacks and wearing coats, which only decrease by 0.6% and 1%.

## 6. Conclusions

We propose a gait-image-recognition method that leverages discontinuous human skeleton keypoint extraction, which can effectively improve the influence of various appearance changes, such as clothing and items, on the accuracy of gait recognition. The experimental results on the CSASIA-B data set have verified that using HRNet, which has a good effect in extracting keypoints of the human body at this stage, aggregates the information of independent branches and introduces multi-resolution interaction to improve the extraction accuracy of keypoints of the human body. Using the modified ResGCN network, gait features containing effective identity information can also be extracted, based on discontinuous picture information, which significantly improves recognition compared with other existing methods. The effectiveness of the algorithm proposed in this paper is also verified by testing the CASIA-A gait dataset and the gait images in real scenes.

However, the work we have undertaken to date is far from being enough. The experiment only uses public data sets, which are small in scale, and with simple scenarios; the recognition accuracy needs to be further improved.

Future research directions can be pursued in the following areas:(1)Due to the differences between the gait dataset used in the experiment and the real-world environment, the next step for the work involves testing the algorithm on videos collected in real environments, considering more walking views, and different walking conditions and lighting conditions. This is necessary to achieve pedestrian recognition with algorithm adaptability to the diversity of the database.(2)The various algorithm modules selected in the framework of this paper are continuously being upgraded and iterated with the development of deep learning theory. In future research work, we can consider to replacing the pedestrian object-detection network with a network that provides better detection results. For the human skeleton keypoint-detection network, it is advisable to use a multi-person skeleton keypoint-detection network to address pedestrian gait recognition in crowded places and other scenarios.

## Figures and Tables

**Figure 1 sensors-23-07274-f001:**
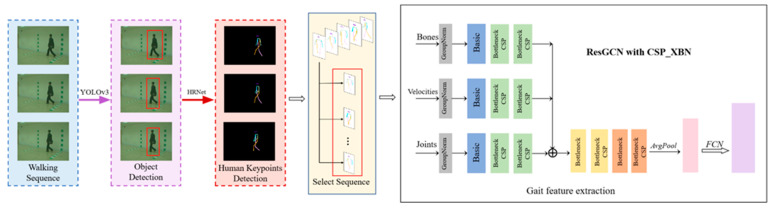
Algorithm block diagram. Starting from the frame-level sequence of video images, YOLOv3 was used to detect pedestrian targets in the images, and body keypoints were extracted for each pedestrian. Discontinuous image sequences were selected for gait feature extraction by the improved ResGCN.

**Figure 2 sensors-23-07274-f002:**
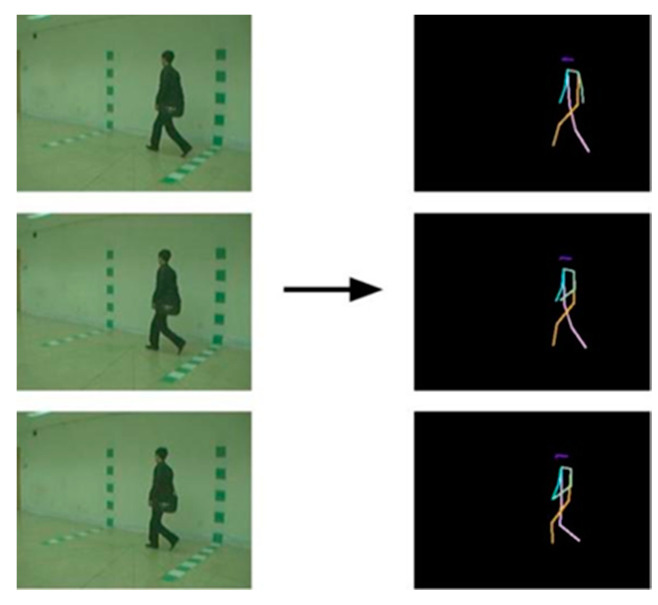
HRNet to CASIA-B 2D skeleton pose extraction.

**Figure 3 sensors-23-07274-f003:**
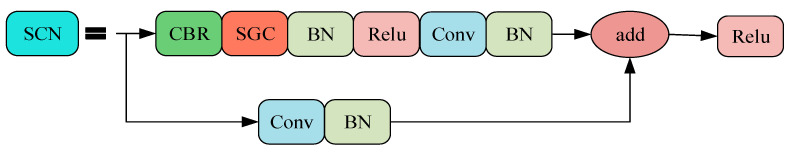
SCN module structure diagram. The SGC in the figure is a spatial graph convolution.

**Figure 4 sensors-23-07274-f004:**
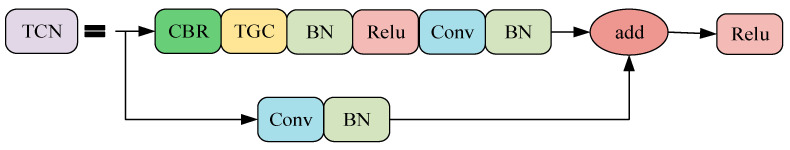
TCN module structure diagram. The TGC in the figure is a temporal graph convolution.

**Figure 5 sensors-23-07274-f005:**
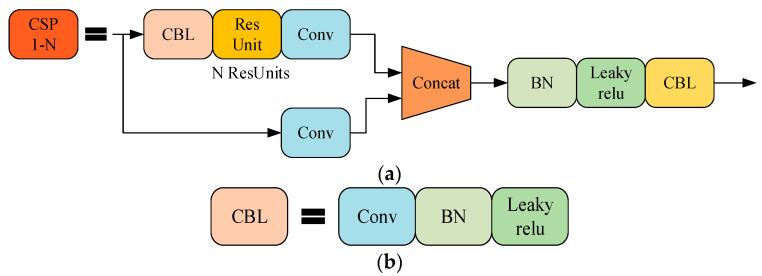
Structure diagram: (**a**) CSP; (**b**) CBL module in CSP.

**Figure 6 sensors-23-07274-f006:**
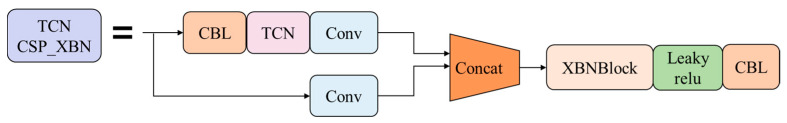
TCN module structure improvement diagram.

**Figure 7 sensors-23-07274-f007:**
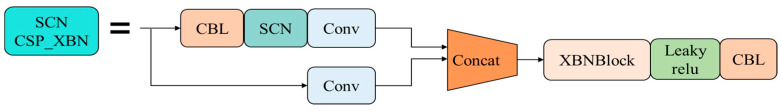
SCN module structure improvement diagram.

**Figure 8 sensors-23-07274-f008:**
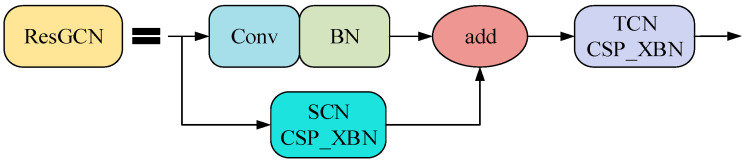
Diagram of the improved ResGCN structure.

**Figure 9 sensors-23-07274-f009:**
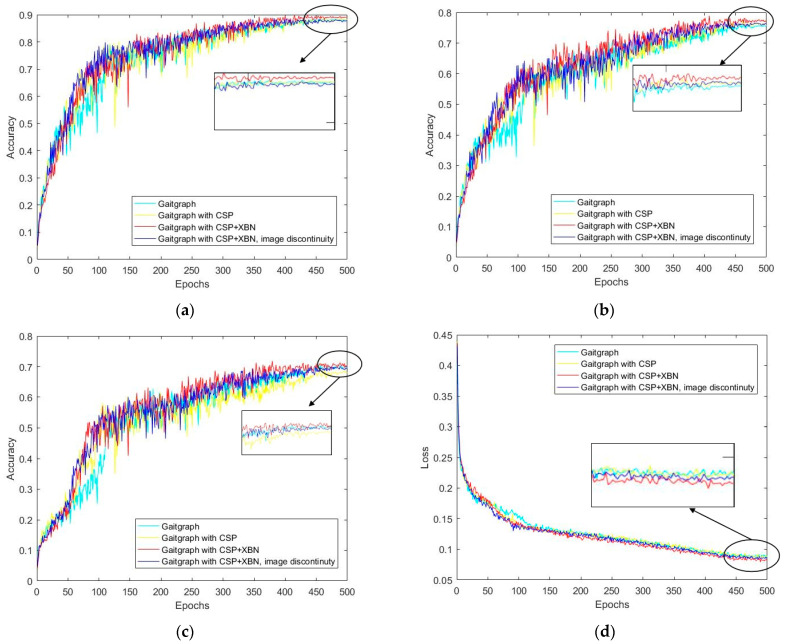
Accuracy of models with different modifications and loss curves: (**a**) NM#5–6; (**b**) BG#1–2; (**c**) CL#1–2. (**d**) Comparison of loss curves.

**Figure 10 sensors-23-07274-f010:**
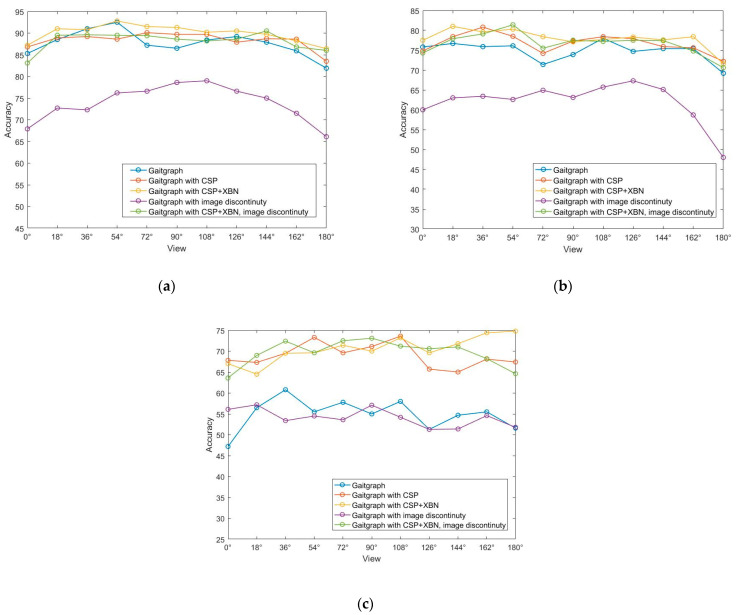
Ablation experiment comparison line chart: (**a**) NM#5–6; (**b**) BG#1–2; (**c**) CL#1–2.

**Figure 11 sensors-23-07274-f011:**
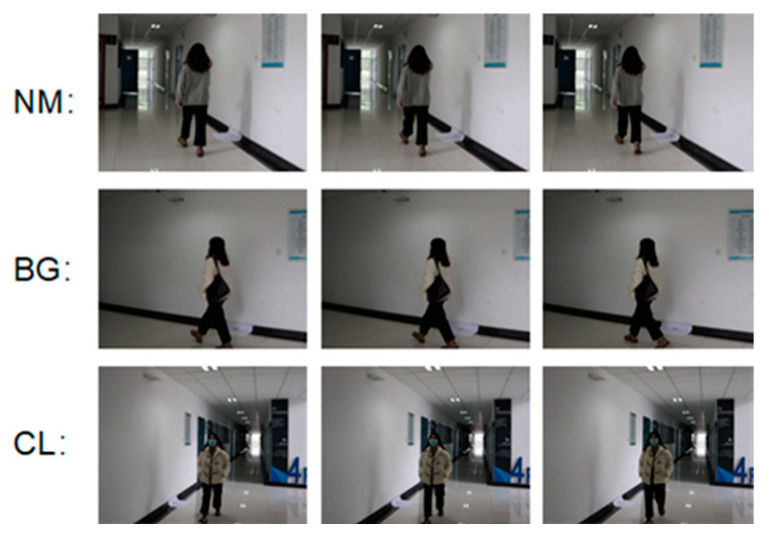
Gait images of three variations in real scenes.

**Table 1 sensors-23-07274-t001:** Overview of the improved *ResGCN-N39-R8* network architecture for poses with 17 joints.

Block	Module
Block 0	GroupNorm
Block 1	Basic
	BottleneckCSP_XBN
	BottleneckCSP_XBN
Block 2	Bottleneck
	BottleneckCSP_XBN
	Bottleneck
	BottleneckCSP_XBN
Block 3	AvgPool2D
Block 0	FCN

**Table 2 sensors-23-07274-t002:** Computer environment for the experiments.

Configuration	Quantity
System	Ubuntu 18.04.5 LTS
CPU	Intel Xeon(R) Gold 5218: 2.30 GHz, 16 cores, 32 threads
Memory	62.6 GB
GPU	NVIDIA GeForce RTX 2080 Ti: 11264 MB, 14000 MHz
Disk	Jesis X760S 512 GB + TOSHIBA MG04ACA4 4.4 TB
OS type	64-bit
Python	version 3.6.9
Nvidia driver	version 450.36.06
CUDA	version 11.0
cuDNN	version 8.0.2
Pytorch	version 1.5.1
OpenCV	version 4.2.0

**Table 3 sensors-23-07274-t003:** Averaged rank-1 accuracies in percent on CASIA-B per probe angle, compared with the baseline model GaitGraph, excluding identical-view cases. Image discontinuity study. Control Condition: interrupt the input sequence at train/test phase. Results are rank-1 accuracies on CASIA-B, averaged in percent.

Gallery NM#1–4		0–180°											Mean
Probe		0°	18°	36°	54°	72°	90°	108°	126°	144°	162°	180°	
NM #5–6	GaitGraph [43]	85.3	88.5	91.0	92.5	87.2	86.5	88.4	89.2	87.9	85.9	81.9	87.7
	GaitGraph [43] with CSP, XBN(Ours)	87.2 (↑1.9)	91.0 (↑2.5)	90.8	92.8 (↑0.3)	91.5 (↑4.3)	91.3 (↑4.8)	90.2 (↑1.8)	90.5 (↑1.3)	89.7 (↑1.8)	88.2 (↑2.3)	86.4 (↑45)	90.0 (↑2.3)
	GaitGraph [43] with CSP, XBN, image discontinuty(Ours)	83.1	89.5 (↑1.0)	89.6	89.5	89.4 (↑2.2)	88.6 (↑2.1)	88.2	88.6	90.5 (↑2.6)	86.8 (↑0.9)	86.0 (↑4.1)	88.2 (↑0.5)
BG #1–2	GaitGraph [43]	75.8	76.7	75.9	76.1	71.4	73.9	78.0	74.7	75.4	75.4	69.2	74.8
	GaitGraph [43] with CSP, XBN(Ours)	77.5 (↑1.7)	81.0 (↑4.3)	79.6 (↑3.7)	80.3 (↑4.2)	78.4 (↑7.0)	77.1 (↑3.2)	77.7	78.3 (↑3.6)	77.6 (↑2.2)	78.4 (↑3.0)	71.6 (↑2.4)	77.9 (↑3.1)
	GaitGraph [43] with CSP, XBN, image discontinuty(Ours)	74.3	77.9 (↑1.2)	79.1 (↑3.2)	81.4 (↑5.3)	75.5 (↑4.1)	77.5 (↑3.6)	77.2	77.5 (↑2.8)	77.4 (↑2.0)	74.8	70.6 (↑1.4)	76.6 (↑1.8)
CL #1–2	GaitGraph [43]	69.6	66.1	68.8	67.2	64.5	62.0	69.5	65.6	65.7	66.1	64.3	66.3
	GaitGraph [43] with CSP, XBN(Ours)	67.0	64.5	69.5 (↑0.7)	69.6 (↑2.4)	71.4 (↑6.9)	70.0 (↑8.0)	73.2 (↑3.7)	69.6 (↑4.0)	71.8(↑6.1)	74.4 (↑8.3)	74.8 (↑10.5)	70.5 (↑4.2)
	GaitGraph [43] with CSP, XBN, image discontinuty(Ours)	63.6	69.0 (↑2.9)	72.4 (↑3.6)	69.6 (↑2.4)	72.5 (↑8.0)	73.1 (↑11.1)	71.2 (↑1.7)	70.6 (↑5.0)	71.0 (↑5.3)	68.2 (↑2.1)	64.6 (↑0.3)	69.6 (↑3.3)

**Table 4 sensors-23-07274-t004:** Ablation experiments conducted on CASIA-B using setting ‘LT’. Results are rank-1 accuracies averaged in percent on all 11 views, excluding identical-view cases.

Model	Probe			Mean
	NM	BG	CL	
GaitGraph [43]	87.7	74.8	66.3	76.3
GaitGraph [43] with CSP	88.3	76.7	68.9	78.0
GaitGraph [43] with CSP and XBNBlock	90.0	77.9	70.5	79.5
GaitGraph [43] with mage discontinuity	73.9	62.0	54.1	63.3
GaitGraph [43] with CSP and XBN, image discontinuity	88.2	76.6	69.6	78.1

**Table 5 sensors-23-07274-t005:** Averaged rank-1 accuracies, in percent on CASIA-B: comparison with other methods.

Type	Model	Probe			Mean
		NM	BG	CL	
appearance-based	GaitNet [30]	91.6	85.7	58.9	78.7
	GaitSet [31]	95.0	87.2	70.4	84.2
	GaitPart [32]	96.2	91.5	78.7	88.8
	3DLocal [34]	97.5	94.3	83.7	91.8
model-based	PoseGait [40]	60.5	39.6	29.8	43.3
	GaitGraph [43]	87.7	74.8	66.3	76.3
	Ours	90.0	77.9	70.5	79.5

**Table 6 sensors-23-07274-t006:** Comparison of related parameters in the improved network experiment with the baseline.

Model	Average Detection Time (ms)	Weight Size (MB)
GaitGraph [43]	0.0996	4.1
GaitGraph [43] with CSP	0.0650	3.8
GaitGraph [43] with CSP and XBNBlock	0.0650	3.8
GaitGraph [43] with CSP and XBN, image discontinuity	0.0660	3.9

**Table 7 sensors-23-07274-t007:** Results are rank-1 accuracies (%) of the proposed method on CASIA-A under 3 probe views, excluding identical-view cases.

Model	Probe			Mean
	0°	45°	90°	
GaitGraph [43]	88.8	87.1	84.2	86.7
GaitGraph [43] with CSP and XBNBlock (Ours)	95.4	92.9	89.6	92.6
GaitGraph [43] with CSP and XBN, image discontinuity (Ours)	91.2	89.2	86.5	89.0

**Table 8 sensors-23-07274-t008:** Results are rank-1 accuracies (%) of the proposed method on real-scene gait images, excluding identical-view cases.

Gallery NM#1–2		View			Mean
Probe		36°	108°	162°	
BG#1–2	Ours	89.2	87.9	85.8	87.6
	Ours with image discontinuity	85.4	83.8	82.9	84.0
CL#1–2	Ours	78.7	76.2	75.8	76.9
	Ours with image discontinuity	75.4	73.7	72.9	74.0

## Data Availability

No new data were created or analysed in this study. Data sharing is not applicable to this article.

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
