# Peer review of "Research Method of Discontinuous-Gait Image Recognition Based on Human Skeleton Keypoint Extraction"

_sensors, 2023, doi:10.3390/s23167274_

Round 1
Reviewer 1 Report
The paper is well-written. However, there are a few suggestions:
1. The researchers have employed the ResGCN technique, which needs to be described in detail.
2. Some of the abbreviations used in the context must be mentioned above for better clarity.
3. The article contains a few grammatical errors that need to be corrected.
The article has a few grammatical errors that require correction.
Author Response
Dear reviewer:
We feel great thanks for your professional review work on our article. As you are concerned, there are several problems that need to be addressed. According to your nice suggestions, we have made extensive corrections to our previous draft, the detailed corrections are listed below.

Reviewer 2 Report
In this paper, the authors propose a novel gait recognition method based on the extraction of human skeleton keypoints from discontinuous images, aiming to counteract the influence of appearance changes on recognition accuracy. They leveraged HRNet for body keypoints extraction and a modified ResGCN network for extracting gait features from discontinuous image data. The method significantly outperformed existing ones, with tests conducted on both the CSASIA-B dataset and real-world gait images. While they acknowledged that the accuracy needs improvement, and the data used were from small, simple public datasets, future work will focus on refining the algorithm for larger, more complex scenarios.
The paper presents a well-structured and comprehensive study on a gait recognition method, demonstrating its efficacy in handling complex real-world challenges such as appearance changes. The authors effectively leverage state-of-the-art tools and techniques, including HRNet and a modified ResGCN network. They also evaluate their proposed model against widely used benchmarks and in realistic scenarios, providing ample evidence of its performance. However, the research does have limitations, mainly pertaining to the limited scale and simplicity of the datasets used. Nevertheless, the authors acknowledge these shortcomings and suggest meaningful avenues for future work, demonstrating an in-depth understanding of the field and the challenges ahead. Overall, the paper contributes insights to the field of gait recognition.
Major comments:
1. The authors have conducted their experiments primarily on public datasets, which they recognize as being small-scale and limited in scope. While such datasets are commonly used, they may not fully encompass the complexities encountered in real-world scenarios. This limitation constrains the generalizability of the study's outcomes.
2. The paper would benefit from a more detailed justification of the chosen methodologies and techniques. Specifically, a clearer explanation for the selection of HRNet and the modified ResGCN would provide enhanced clarity to readers.
3. While the authors reference several existing algorithms and models, a more comprehensive comparison with current state-of-the-art models would be useful. The paper currently leaves readers uncertain about how the proposed method compares with the latest advances in the field.
4. The authors have included real-world tests, which is commendable. However, the scope of these tests is quite limited, encompassing only six volunteers. Broader real-world testing could offer more insights into the model's performance.
5. The "Future Work" section could include a more detailed roadmap, suggesting potential exploration of alternative techniques, models, and applications. This addition would provide readers with a better understanding of possible improvements and future research directions.
6. The tables and figures in the paper would benefit from more detailed descriptions or captions to ensure they can be understood without reference to the main text. Furthermore, Table 3 contains some unexplained Chinese words which may confuse readers.
7. The primary metric used for model evaluation in this paper is accuracy. Depending on the task, however, employing other metrics such as precision, recall, F1-score, or the area under the ROC curve could provide a more holistic performance evaluation.
8. Testing appears to be confined to specific walking states and angles. A more diverse range of scenarios and conditions could result in a more rigorous assessment of the model's robustness and versatility.
9. The paper does not provide information on the computational cost associated with the proposed model. For practical applications, understanding the model's computational demands and its feasibility for real-world implementation is crucial.
10. The authors do not mention whether they applied cross-validation to the results.
11. An error analysis could be beneficial to understand where and why the model errs, providing valuable insights for future model improvements.
12. The paper omits a discussion on the impact of varied lighting conditions on the proposed model. Given that lighting changes can significantly affect image-based models, this could be a considerable limitation in real-world applications.
Some phrases and word choices might benefit from minor improvements for clarity and academic tone. For instance, phrases like "We propose a discontinuous gait image recognition research method based on human skeleton keypoint extraction" might be better worded as "We propose a gait image recognition method that leverages discontinuous human skeleton keypoint extraction." Such refinements could further enhance the readability and clarity of the text. Moreover, ensuring consistency in terminology, the correct use of prepositions, and maintaining subject-verb agreement would help improve the overall quality of the paper.
Author Response
Dear reviewer,
Thank you very much for your comments and professional advice. These opinions help to improve academic rigor of our paper. Based on your suggestion and request, we have made corrected modifications on the revised manuscript. The detailed corrections are listed in PDF.

Round 2
Reviewer 2 Report
Thanks for addressing my comments.
Author Response
Dear reviewer,
Thank you for accepting our response, and we sincerely appreciate the professional advice you have provided once again. Based on your comments and suggestions, we have made meticulous revisions to the manuscript, further improving its content. The modified sections are highlighted in yellow in the updated manuscript. More revision details are listed in the accompanying PDF for your reference.
